# From voxels to pixels and back: Self-supervision in natural-image reconstruction from fMRI

**Roman Beliy**\*
Dept. of Computer Science and Applied Math
The Weizmann Institute of Science
76100 Rehovot, Israel
`roman.beliy@weizmann.ac.il`

**Guy Gaziv**\*
Dept. of Computer Science and Applied Math
The Weizmann Institute of Science
76100 Rehovot, Israel
`guy.gaziv@weizmann.ac.il`

**Assaf Hoogi**
Dept. of Computer Science and Applied Math
The Weizmann Institute of Science
76100 Rehovot, Israel
`assaf.hoogi@weizmann.ac.il`

**Francesca Strappini**
Dept. of Neurobiology
The Weizmann Institute of Science
76100 Rehovot, Israel
`francescastrappini@gmail.com`

**Tal Golan**
Zuckerman Institute
Columbia University
10027 New York, NY, USA
`tal.golan@columbia.edu`

**Michal Irani**
Dept. of Computer Science and Applied Math
The Weizmann Institute of Science
76100 Rehovot, Israel
`michal.irani@weizmann.ac.il`

## Abstract

Reconstructing observed images from fMRI brain recordings is challenging. Unfortunately, acquiring sufficient "labeled" pairs of {Image, fMRI} (i.e., images with their corresponding fMRI responses) to span the huge space of natural images is prohibitive for many reasons. We present a novel approach which, in addition to the scarce labeled data (training pairs), allows to train fMRI-to-image reconstruction networks also on "unlabeled" data (i.e., images without fMRI recording, and fMRI recording without images). The proposed model utilizes both an Encoder network (image-to-fMRI) and a Decoder network (fMRI-to-image). Concatenating these two networks back-to-back (Encoder-Decoder & Decoder-Encoder) allows augmenting the training with both types of unlabeled data. Importantly, it **allows training on the unlabeled test-fMRI data**. This self-supervision adapts the reconstruction network to the new input test-data, despite its deviation from the statistics of the scarce training data.

*Project Website:* http://www.wisdom.weizmann.ac.il/~vision/ssfmri2im/

## 1 Introduction

Developing a method for high-quality reconstruction of seen images from the corresponding brain activity is an important milestone towards decoding the contents of dreams and mental imagery (Fig 1a). In this task, one attempts to solve for the mapping between fMRI recordings and their corresponding natural images, using many "labeled" {Image, fMRI} pairs (i.e., images and their corresponding fMRI responses). A good fMRI-to-image decoder is one that will generalize well to reconstruction of new never-before-seen images from new fMRI recordings (we refer to these as

---

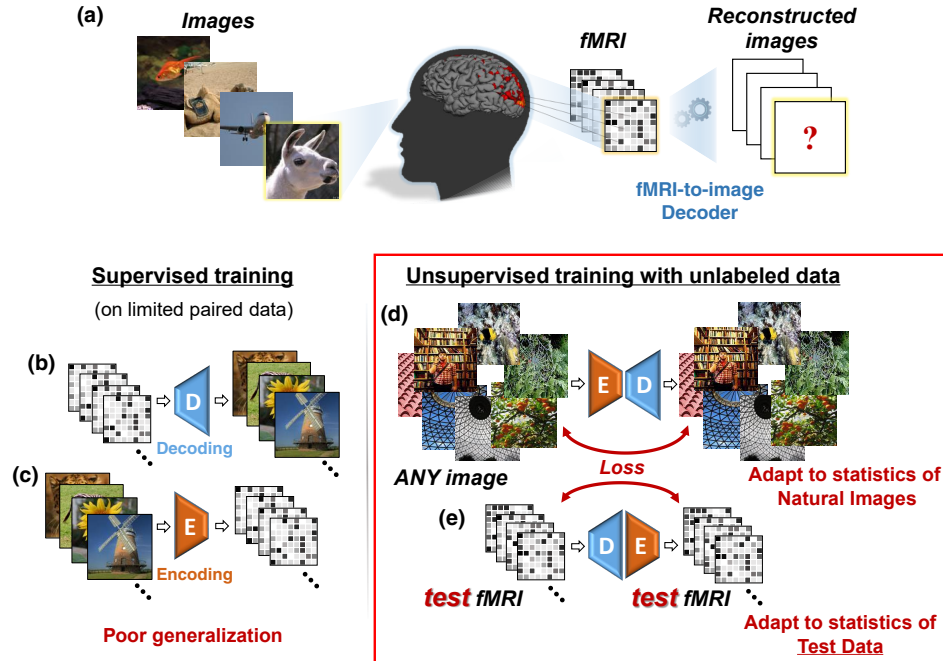

Figure 1: **Our proposed method.** *(a) The task: reconstructing images from evoked brain activity, inferred from fMRI responses. (b), (c) Supervised training for decoding (b) and encoding (c) using limited training pairs. This gives rise to poor generalization. (d), (e) Illustration of our added self-supervision, which enables training on "unlabeled images" (any natural image with no fMRI recording – (d)), and on the "unlabeled fMRI" (fMRI data without any corresponding images – (e)). In particular, the latter allows adapting the decoder to the statistics of the target test-fMRI despite not having any information about their corresponding images.*

"test-data" or "test-fMRI"). *However, the lack in "labeled" training data limits the generalization power of today's fMRI decoders.* Acquiring a large number of labeled pairs {Image, fMRI} is prohibitive, due to the limited time a human can spend in an MRI scanner. As a result, most datasets are limited to a few thousands of such pairs. Such limited samples cannot span the huge space of natural images, nor the space of their fMRI recordings. Moreover, the poor spatio-temporal resolution of fMRI signals, as well as their low Signal-to-Noise Ratio (SNR), reduce the reliability of the already scarce labeled training data. Lastly, the train-set and test-set of the fMRI data often *differ* in their statistical properties, specifically in their SNR. This SNR discrepancy is due to averaging a different number of repeated recordings per image (typical of many fMRI datasets). It therefore introduces an additional challenge of 'domain transfer/adaptation', which makes generalization even harder, and affects the performance of current decoding methods.

**Prior work in image reconstruction from fMRI.** The task of reconstructing a visual stimulus from fMRI has been approached by a number of methods which can broadly be classified into three families: (i) Linear regression between fMRI data and handcrafted image-features (e.g., Gabor wavelets) [1, 2, 3], (ii) Linear regression between fMRI data and Deep (CNN-based) image-features (e.g., using pretrained AlexNet) [4, 5, 6], and (iii) End-to-end Deep Learning [7, 8, 9, 10].

The first two regression-based methods compute a linear model that relates fMRI voxels to image feature representation. This can be done by either linearly predicting each voxel's responses from the image features [1, 2, 3, 5], or by linearly mapping voxel responses to image features from which the image can be easily recovered [4, 6]. The feature representation is chosen such that it closely mimics the neural activity in the visual cortex, with the hope that a simple model (like linear regression) will suffice to capture the remaining mapping. Methods in the second category benefited from utilizing data-driven learned features from leading CNN models trained for natural image classification [5, 11, 12, 13, 7, 14, 15]. The last category refers to recent attempts to train high-complexity deep models which directly decode an fMRI recording into its corresponding image stimulus. To our best knowledge, methods [6] and [8] are the current state-of-the-art in this field.

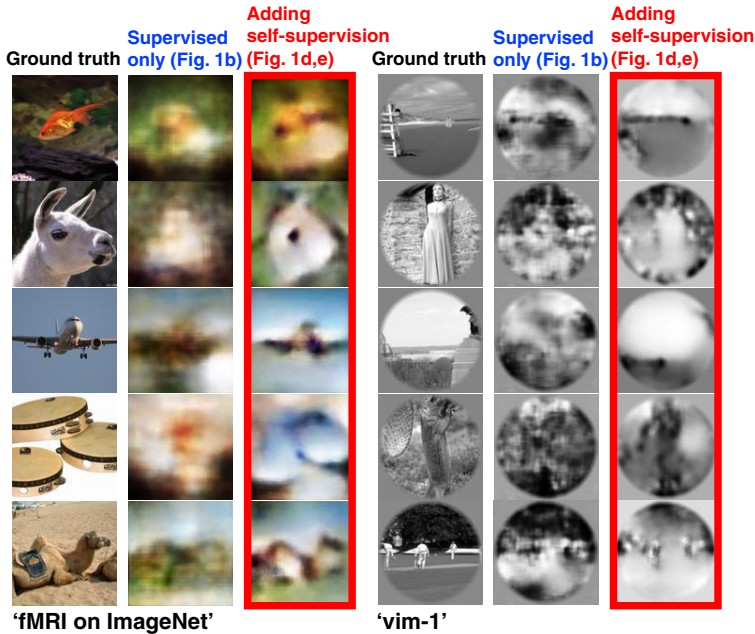

**Ground truth** / **Supervised only (Fig. 1b)** / **Adding self-supervision (Fig. 1d,e)** — **'fMRI on ImageNet'**

**Ground truth** / **Supervised only (Fig. 1b)** / **Adding self-supervision (Fig. 1d,e)** — **'vim-1'**

Figure 2: **Adding un-supervised training on unlabeled data improves reconstruction.**
*(Left column): the images presented to the human.*
*(Middle column): Reconstruction using the training pairs only (Fig 1b).*
*(Right column): Reconstruction when adding unsupervised training on unlabeled data (Fig 1d,e). Example results are shown for two fMRI datasets: 'fMRI on ImageNet' [16] and 'vim-1' [1].*

All these methods are **supervised** (i.e., train their decoder on pairs of {Image, fMRI}), hence suffer from the very limited training data. Purely supervised models are prone to poor generalization to new test-data (fMRI of new images). To overcome this problem, recent methods [6, 8, 9, 10] constrain the reconstructed image to have natural-image statistics by introducing Generative Adversarial Networks (GANs) into their decoder. These methods gave leap advancement in reconstruction quality from fMRI, and tend to produce natural-looking images. Nevertheless, despite their pleasant natural appearance, their reconstructed images are often *unfaithful* to the actual images underlying the test-fMRI (see Fig 5a).

We present a new approach to overcome the inherent lack of training data and the discrepancy between the train/test statistics, by introducing *self-supervision using unlabeled data*. Our approach is illustrated in Fig 1. We train two types of networks: an Encoder $E$, to map natural images to their corresponding fMRI response, and a Decoder $D$, to map fMRI recordings to their corresponding images. Concatenating those two networks back-to-back, E-D, yields a combined network whose input and output are the same image (Fig. 1d). ***This allows for unsupervised training on unlabeled images*** (i.e., images without fMRI recordings, e.g., 50,000 randomly sampled natural images in our experiments). Such self-supervision adapts the network to the statistics of never-before-seen images. Moreover, concatenating our two networks the other way around, D-E, yields a combined network with the same shared weights as E-D, but whose input and output are now an fMRI signal (Fig. 1e). ***This allows unsupervised training on unlabeled fMRI***. Specifically, those unlabeled fMRI samples can be legitimately drawn from the test-fMRI cohort, while their corresponding images ("test-images") are excluded from training (Fig 1e).

**Training on these unlabeled test-fMRI (without their images) is a key feature of our method**: it enables to adapt the network to the statistics of the new (unlabeled) test-data. Learning the statistics of the test-fMRI directly addresses the lack in labeled training data and the train/test statistics discrepancy. Note that our "training on test data", which may seem "illegal" at first sight, is in fact valid. It refers only to training on ***unlabeled samples from the Decoder's input space (test-fMRI)***, whereas the *test-images* (the "labels") are never used at any stage of the training.

Fig. 2 exemplifies the power of adding unsupervised training on unlabeled data. Notably, we found the unsupervised training on the unlabeled test-fMRI (Fig 1e) to provide the greatest boost in performance.

Unsupervised training on unlabeled natural images was also recently proposed in [8], where they used these images to produce additional surrogate fMRI-data to train their model. This, however, does not help to adapt the network to the statistics of the new test-fMRI. To the best of our knowledge, we are the first to provide an approach for adding self-supervised training on unlabeled fMRI. This self-

supervision provides an improvement in decoding of never-before-seen images from brain activity, despite the very limited training data.

Insufficient training data and domain-adaptation [17] are the focus of many recent machine learning works, including transfer learning [18], unsupervised and self-supervised learning [19, 20, 21, 22], semi-supervised and transductive learning [23, 24, 25, 26]. Nevertheless, these approaches often assume that there is sufficient labeled data in the 'source domain', and that the problem is the generalization to the unlabeled 'target domain'. This, however, is not the case here. There is too little data (both labeled or unlabeled) to work with in this challenge. The approach we took is inspired by the recent advances in "Deep-Internal-Learning" [27, 28, 29], which train an *image-specific network*, at test time, on the test-image alone, without requiring any prior training examples. Our approach combines ideas from Internal-Learning with supervised-learning, to get the best of both worlds.

Our contributions are therefore several-fold:

- We propose a new approach to handle the inherent lack in Image-fMRI training data.
- To the best of our knowledge, we are the first to suggest an approach for self-supervised training on unlabeled fMRI data (with no images), and in particular, on the test-fMRI data.
- We demonstrate the power and versatility of our approach by applying it (with the same architecture and same hyperparameters) to two very different fMRI datasets. We achieve competitive results in image reconstruction on both datasets ('fMRI on ImageNet' [16], and 'vim-1' [1]). Most methods, including those we compared our results with, are adapted to only one dataset.

## 2 Method overview

Our training consists of two phases which are illustrated in Fig 3. In the first phase, we apply supervised training of the Encoder $E$ alone. We train it to predict the fMRI responses of input images using the image-fMRI training pairs (Fig 3a). In the second phase, we use the pretrained Encoder (from the first phase) and train the Decoder $D$, keeping the weights of $E$ fixed. $D$ is trained jointly using both the labeled and the unlabeled data, simultaneously. Each training batch consists of three types of training data: (i) labeled image-fMRI pairs from the training set (Fig 1b), (ii) unlabeled natural images (Fig 1d), and (iii) unlabeled fMRI (Fig 1e).

Specifically, we draw the *unlabeled images* from a large *external* database of 50K ImageNet images, which is disjoint to the considered image-fMRI dataset. This promotes adaptation of the Decoder to the statistics of natural images. The *unlabeled fMRI data* is drawn from the unlabeled test-fMRI cohort (without any test-images, i.e., without their "labels"). This promotes adaptation of the Decoder to the statistics of the fMRI test data. Once completed, inference of test stimuli is carried out by feeding-forward the test-fMRI through the trained Decoder.

Note that our "training on test data", which may seem "illegal" at first sight, is in fact valid. It refers only to training on **unlabeled samples from the Decoder's input space** *(test-fMRI)*, whereas the *test-images* (the "labels") are never used at any stage of the training.

The motivation for using two training phases is to allow the Encoder to converge at the first phase, and then serve as strong guidance for the more severely ill-posed decoding task, which is the focus of the second phase. The weights of the Encoder are kept fixed during the Decoder training, to ensure that the Encoder's output representation does not diverge from predicting fMRI responses by the unsupervised training objectives 1d,e.

We next describe each phase in more detail. We start by supervised training of the Encoder.

### 2.1 The Encoder $E$ (Images → fMRI)

The training of the Encoder is illustrated in Fig. 3a. Let $\hat{r} = E(s)$ denote the encoded fMRI response from image, $s$, by Encoder $E$. We define fMRI loss by a convex combination of mean square error and cosine proximity with respect to the ground truth fMRI, $r$. The **fMRI loss** is defined as:

$$\mathcal{L}_r(\hat{r}, r) = \alpha \|\hat{r} - r\|_2 - (1 - \alpha) \cos(\angle(\hat{r}, r)),\tag{1}$$

where $\alpha$ is a hyperparameter set empirically (see Implementation Details). We use this loss for training the Encoder $E$. However, this loss is also used to define the Decoder-Encoder loss (unlabeled fMRI) on which we detail later.

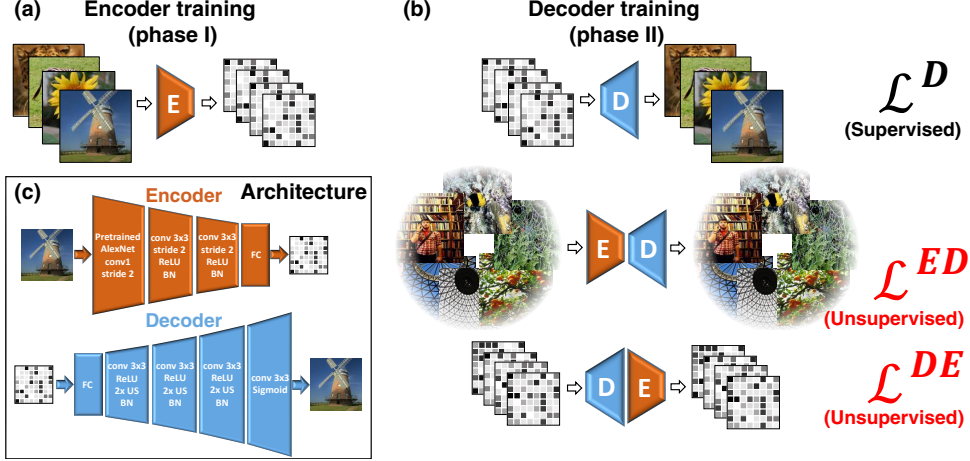

Figure 3: **Training phases & Architecture.** *(a) The first training phase: Supervised training of the Encoder with {Image, fMRI} pairs. (b) Second phase: Training the Decoder simultaneously with 3 types of data: {Image, fMRI} pairs (supervised examples), unlabeled natural images (self-supervision), and unlabeled test-fMRI (self-supevision). Note that the test-images are never used for training. The pretrained Encoder from the first training phase is kept fixed in the second phase. (c) Encoder and Decoder architectures. BN, US, and ReLU stand for batch normalization, up-sampling, and rectified linear unit, respectively.*

Notably, in the considered fMRI datasets, the subjects who participated in the experiments were instructed to fixate at the center of the images. Nevertheless, eye movements were not recorded during the scans thus the fixation performance is not known. To accommodate the center-fixation uncertainty, we introduced random shifts of the input images during Encoder training. This resulted in a substantial improvement in the Encoder performance and subsequently in the image reconstruction quality. Upon completion of Encoder training, we transition to training the Decoder together with the fixed Encoder.

## 2.2 The Decoder $D$ (fMRI → Images)

The training loss of our Decoder consists of three main losses illustrated in Fig. 3b:

$$\mathcal{L}^D + \mathcal{L}^{ED} + \mathcal{L}^{DE}. \tag{2}$$

$\mathcal{L}^D$ is a supervised loss on training pairs of image-fMRI. $\mathcal{L}^{ED}$ and $\mathcal{L}^{DE}$ are unsupervised losses on unlabeled images (without fMRIs) and unlabeled fMRIs (without images). All 3 components of the loss are normalized to have the same order of magnitude (all in the range $[0, 1]$, with equal weights), to guarantee that the total loss is not dominated by any individual component. We found our reconstruction results to be relatively insensitive to the exact balancing between the three components (see Supplementary-Material).

We next detail each component of the loss.

**Decoder Supervised Training** is illustrated in Fig. 1b. Given training pairs $\{(r, s)\}$, the supervised loss $\mathcal{L}^D$ is applied on the decoded stimulus, $\hat{s} = D(r)$, and is defined via the image reconstruction objective, $\mathcal{L}_s$, as

$$\mathcal{L}^D = \mathcal{L}_s(\hat{s}, s).$$

$\mathcal{L}_s$ consists of losses on image RGB values, $\mathcal{L}_{RGB}$, and its features, $\mathcal{L}_{features}$. We denote the features extracted from an image, $s$, by $\varphi(s)$, and chose $\varphi$ to be pretrained a feature-extractor. Specifically we used activations from the first and the second convolutional layers of VGG19 [30]. The Image loss for a reconstructed image $\hat{s}$ reads:

$$\mathcal{L}_s(\hat{s}, s) = \mathcal{L}_{RGB}(\hat{s}, s) + \mathcal{L}_{features}(\hat{s}, s) + \mathcal{R}(\hat{s}) \tag{3}$$

$$\mathcal{L}_{RGB}(\hat{s}, s) \propto \|\hat{s} - s\|_1, \quad \mathcal{L}_{features}(\hat{s}, s) \propto \|\varphi(\hat{s}) - \varphi(s)\|_2, \quad \mathcal{R}(\hat{s}) \propto TV(\hat{s}). \tag{4}$$

The last term corresponds to total variation (TV) regularization of the reconstructed image, $\hat{s} = D(r)$. In addition to defining the Decoder supervised loss, the Image loss is also used to define the Encoder-Decoder loss (unlabeled images) explained later.

We now detail on the crux of our method: Unsupervised training with unlabeled data.

**Encoder-Decoder training on unlabeled Natural Images** is illustrated in Fig. 1d. This objective enables to train on any desired unlabeled image, well beyond the 1200 images included in the main fMRI dataset. To train on images without corresponding fMRI responses, we map images to themselves through Encoder-Decoder transformation,

$$s \mapsto \hat{s}_{ED} = D\left(E\left(s\right)\right).$$

The unsupervised component $\mathcal{L}^{ED}$ of the loss in Eq 2 on unlabeled images, $s$, reads:

$$\mathcal{L}^{ED} = \mathcal{L}_s\left(\hat{s}_{ED}, s\right),$$

where $\mathcal{L}_s$ is Image loss defined in Eq 3.

**Decoder-Encoder training on unlabeled test fMRI** is illustrated in Fig. 1e. Adding this objective greatly improved our reconstruction quality compared to training on paired samples only. To train on fMRI data without corresponding images, we map an fMRI response to itself through Decoder-Encoder transformation:

$$r \mapsto \hat{r}_{DE} = E\left(D\left(r\right)\right).$$

This yields the following unsupervised component $\mathcal{L}^{DE}$ of the loss in Eq 2 on unlabeled fMRI responses $r$:

$$\mathcal{L}^{DE} = \mathcal{L}_r\left(\hat{r}_{DE}, r\right),$$

where $\mathcal{L}_r$ is fMRI loss defined in Eq 1.

Importantly, the fMRI samples which we used here were drawn from the test cohort. This enables to adapt the Decoder to the statistics of the test-fMRI data (which we want to decode). The same test-fMRI data is subsequently used at inference.

## 2.3 Implementation details

We focused on 112x112 RGB or grayscale image reconstruction (depending on the dataset), although our method works well also on other resolutions.

**Architectures** of the Encoder and the Decoder are illustrated in Fig. 3c. For the Decoder we used a fully connected layer to transform and reshape the vector-form fMRI input into 64 feature maps with spatial resolution 14x14. This representation is then followed by three blocks, each consists of: (i) 3x3 convolution with unity stride, 64 channels, and ReLU activation, (ii) x2 up-sampling, and (iii) batch normalization. To yield the output image we finally performed an additional convolution, similar to the preceding ones, but with three channels to represent colors, and a sigmoid activation to keep the output values in the 0-1 range. We used Glorot-normal[31] to initialize the weights. The design of the Encoder consists of feature extraction using pretrained AlexNet conv1 weights, followed by batch normalization. The next operations include three blocks of 3x3 convolution with 32 channels, ReLU activation with stride 2, and batch normalization. Lastly, we use a fully connected layer to bring the output to voxel space. We initialized the weights using Glorot normal initializer.

**Hyperparameter tuning.** We trained the Encoder with $\alpha = 0.9$ using SGD optimizer for 80 epochs with an initial learning rate of 0.1, with a predefined learning rate scheduler. During Decoder training with supervised and unsupervised objectives, each training batch contained 60% paired data (supervised training), 20% unlabeled natural images (without fMRI), and 20% unlabeled test-fMRI (without images). We trained the Decoder for 150 epochs using Adam optimizer with an initial learning rate of 1e-3, and 80% learning rate drop after every 30 epochs.

**Runtime.** Our system completes the two-stage training within approximately 15 min using a single Tesla V100 GPU while inference (decoding) is performed in real time.

**Experimental datasets.** We experimented with two publicly available (and very different) benchmark fMRI datasets, using the same architectures and hyperparameters: (i) fMRI on ImageNet [16], and (ii) vim-1 [1]. These datasets provide fMRI recordings paired with their corresponding underlying images. Subjects were instructed to fixate at a cross located at center of the presented images. **'fMRI on ImageNet'** comprises 1250 distinct ImageNet images drawn from 200 selected categories. The train- and test-fMRI data consist of 1 and 35 (repeated recordings) per presented stimulus, respectively. Fifty image categories provided the fifty test images, one from each category. The remaining 1200

were defined as train set (with only one fMRI recording). We considered approximately 4500 voxels from the visual cortex provided by the authors of [16]. **'vim-1'** comprises 1870 distinct grayscale images. fMRI was recorded (i) twice for 1750 images and defined the training data, and (ii) 13 times for the remaining 120 images, defining the test data.

We screened approximately 8500 out of the 50K recorded voxels by their SNR. We used additional 50K **unlabeled natural images from ImageNet** [32] validation data for the unsupervised training on unlabeled images (Encoder-Decoder objective, Fig. 1d). We verified that the images in our additional unlabeled external dataset, are distinct from those in the "fMRI on ImageNet".

**Performance evaluation.** The reconstruction quality of images from fMRI was assessed both visually and objectively, and was compared with the two leading methods [6, 8] (Fig. 5). The similarity measure was based on correlating pixel values between the reconstructed image and an original image. However, the absolute correlation value on its own is meaningless, and cannot be compared across different types of reconstructions, because of its sensitivity to variations in edge intensity, edge misalignments, etc. For example, when the edges of the reconstructed image are not aligned with those of the ground-truth image (as in the reconstructed white goat in Fig. 4d), standard image-to-image similarity measures will favor a blurrier version of the reconstructed image (e.g., the goat in Fig. 4c) over a sharp one (the goat in Fig. 4d). To alleviate this inherent bias, we used an objective image-reconstruction quality measure by computing its 'correct-identification rate' in a multi-image identification task (as proposed in [6]). The correlation measure, while not ideal, would still produce higher correlation value with the ground-truth image, than with other *random sharp images*. For each reconstructed image, the task is to identify its ground truth image among $n$ candidate images ($n$= 2, 5 or 10), one being the true ground truth, while the rest were randomly selected. This identification was based on the same correlation measure (between the reconstructed image and each candidate image). The candidate image which scored the max Pearson correlation was determined to be the identified 'ground truth'.

Because of the randomness in our training process we repeated the analysis multiple times and averaged over the reconstructed images: 20 runs for 'fMRI on ImageNet', and 10 runs for 'vim-1'.

# 3   Experimental results

Fig. 2 shows our results with the proposed method, which includes the combined supervised and unsupervised training. These results (in red frames) are contrasted with the results obtainable when using supervised training only (with the 1200 labeled training pairs). All the displayed images were reconstructed from the test-fMRI. The red-framed images show many faithfully-reconstructed shapes, textures, and colors, which depict recognizable scenes and objects. In contrast, using the supervised objective alone led to reconstructions that were considerably less faithful and recognizable (middle columns in Fig. 2). The reconstructions of the entire test cohort (50 images) can be found in the Supplementary-Material.

**Ablation study of the method components**

Fig. 4 shows an analysis of the merit of our unsupervised training. Our complete method, which includes training on unlabeled images and on unlabeled test-fMRI is compared against three baselines:

(i) A purely supervised approach (Fig. 1b), relying only on image-fMRI pairs (Fig. 4b). (ii) Adding also unsupervised training on many external unlabeled images (Fig. 4c). Our results suggest a discernible albeit moderate improvement due to this objective. (iii) Adding also unsupervised training on unlabeled fMRI data from the test-fMRI cohort (Fig. 4d). This provides a dramatic improvement in the results, However, **excluding the single unlabeled target-fMRI** of the specific test-image (Fig. 4e) leads to a marked degradation in reconstruction quality. This indicates the importance of adapting our model to the actual test-fMRI which is designated for inference.

We evaluated the reconstruction quality of each component of our method using n-way identification-task based on pixel-level similarity of the reconstructed images and candidate ground truth images. This evaluation confirmed the qualitative trend of better reconstruction by the full method compared to the ablated versions 1b,d,e, at varying number of candidate ground-truth images: $n = 2, 5, 10$ (see Performance Evaluation). For a 2-way identification task (detecting the source of a reconstructed image among two candidate images) we report average scores of 80.1% for supervised training-only, 83.2% when adding the training on additional unlabeled images, and 85.3% for the full method. The identification accuracy dropped to 84.1% when the target test-fMRI was excluded from training.

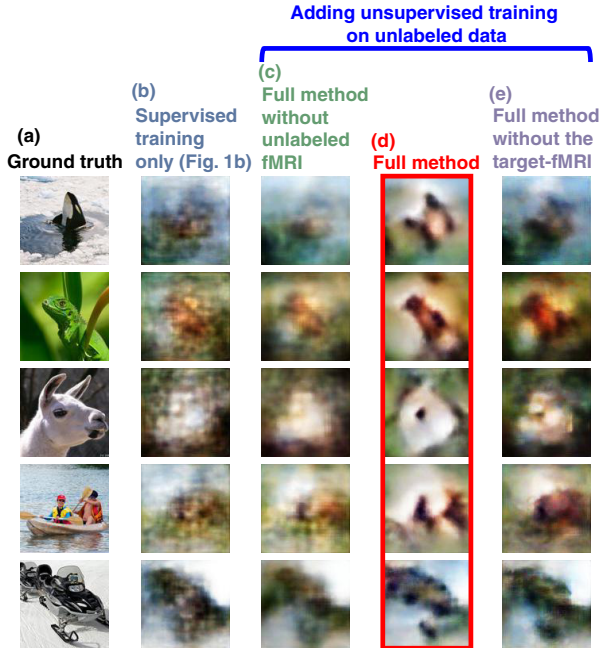

Adding unsupervised training on unlabeled data

**(b)** Supervised training only (Fig. 1b)

**(c)** Full method without unlabeled fMRI

**(d)** Full method

**(e)** Full method without the target-fMRI

**(a)** Ground truth

Figure 4: **Ablation of method components (visual & quantitative).**
*(b) Supervised training on {Image, fMRI} pairs (Fig.1b). (c) Adding unsupervised training, but only on unlabeled images (Fig.1d). (d) Adding also unsupervised training on the unlabeled test-fMRI (Fig.1e). (e) Same as (d), but now the single unlabeled test-fMRI underlying the test-image (i.e., the "target-fMRI") is omitted from the self-supervision on the test-fMRI cohort.*

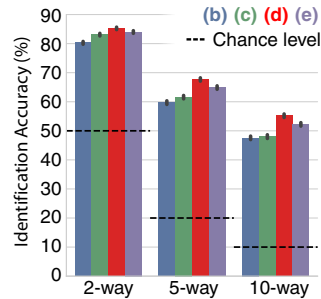

A major factor underlying the performance leap when training on unlabeled test-fMRI (incorporating $\mathcal{L}^{DE}$) was the repeat-count of test-fMRI recordings. Averaging over multiple repeats results in fMRI samples with higher SNR, compared to those in the training data which only have a single repeat. Our ablation studies showed that reconstruction improves as the number of averaged test-fMRI repeats increases (see Supplementary-Material). This shows that our D-E architecture exploits the better SNR of the test-fMRI and adapts $D$ to the statistics of the test-fMRI, which deviates from the statistics (SNR) of the training data.

**Comparison with state-of-the-art methods**

We compared our results both visually and quantitatively against the two leading methods: Shen et al. [6]) and St-Yves et al. [8] – each on its relevant dataset.

***Visual comparison.*** Fig 5a,b compares the results of our method with the corresponding ones proposed in [6, 8]. Each of these methods focused on one specific fMRI dataset, either 'fMRI on ImageNet' [16] or 'vim-1' [1]. Both methods used GANs as natural image priors to increase their generalization power when having very limited training data, resulting in natural-looking images in some cases however substantially deviant from the actual images underlying the fMRI (Shen et al. [6]) and/or low quality (St-Yves et al. [8]). Our method seems to better reconstruct shapes, details and global layout in the reconstructed images than [6, 8]. This is supported visually and numerically.

***Quantitative comparison.*** We report quantitative objective comparisons of the reconstructed images by our method and those by [6, 8] in Fig. 5c,d. These panels show the correct-identification rate (within a method) for n-way classification tasks for $n = 2, 5, 10$ (see Performance Evaluation). We evaluate our method and two variants of the method of [6] on the 'fMRI on ImageNet' benchmark dataset (Fig. 5c). Our method scored 85.3% mean identification accuracy, competing favorably against both variants of [6] by a margin of at least 5% across all task difficulty levels ($n = 2, 5, 10$). We repeated the analysis for 'vim-1' fMRI dataset (Fig. 5d), where our method scored accuracy of 70.5% (for $n = 2$), outperforming the method from [8] by at least 3% across difficulty levels. Taken together, our method competes favorably and slightly outperform state-of-the-art methods. This advantage holds at least with respect to the two considered datasets, and is robust to varying difficulty levels of the identification task.

## Conclusion

This work highlights the importance of self-supervised training on *unlabeled* input test data. This addresses the inherent lack in labeled (supervised) training data, and the discrepancy between the

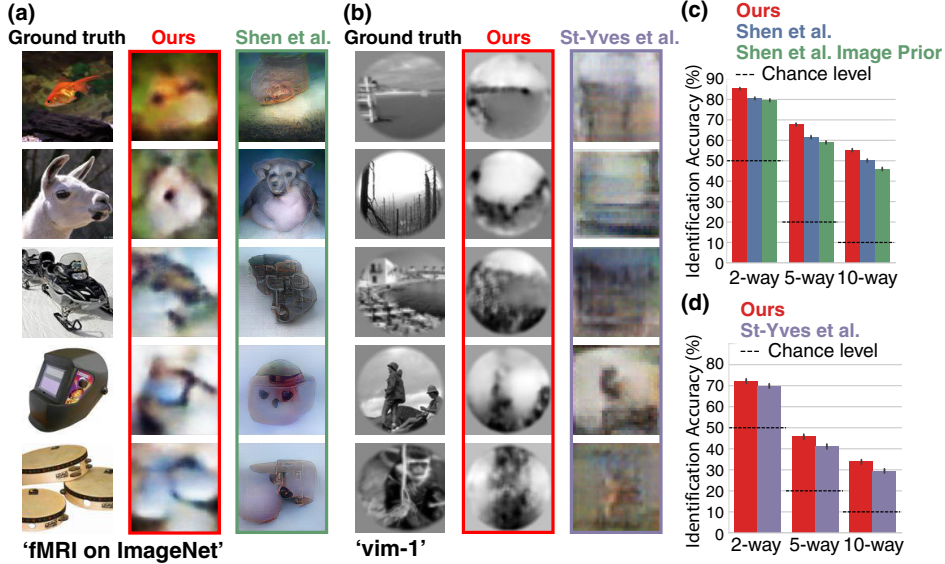

Figure 5: **Comparison with state-of-the-art methods.** *(a), (b) Visual comparison with [6, 8] – each compared on its relevant dataset. Our method reconstructs shapes, details and global layout in images better than the leading methods. (c), (d) Quantitative comparisons of identification accuracy (per method) in an n-way identification task (see text for details).* $95\%$ *Confidence Intervals shown on charts.*

statistics of the train/test data. Our experiments show that self-supervised training on unlabeled test-fMRI (without using any test-images for training) has a dramatic effect on the decoded images. It further has a stronger effect than self-supervised training on unlabeled natural images only. Particularly, including self-supervision on the *target*-fMRI shows the highest impact on the reconstruction of the corresponding target image. These suggest the importance of *adapting the network to the statistics of the input test data*.

While image reconstruction can eventually become a strong neuroscientific tool, this is not the focus of the current work. This work highlights a new learning method, which is exemplified on a difficult neuroscientific problem, but is not limited to it. The characteristics of the fMRI-inference problem are common to other ill-posed learning tasks where labeled training data is scarce, while high generalization power is desired. Adapting to the statistics of the target test data may be useful for promoting generalization for those other problem areas as well.

## Acknowledgments

This project has received funding from the European Research Council (**ERC**) under the European Union's Horizon 2020 research and innovation programme (grant agreement No 788535).

## Author Contributions

R.B. and G.G. designed the experiments. R.B. implemented the network and conducted the image-reconstruction experiments. G.G. designed and wrote the paper, and analyzed the fMRI data. A.H. conducted reconstruction quality analyses. F.S. and T.G. provided guidance on fMRI preprocessing. M.I. conceived the original idea and supervised the project. All authors discussed the results and commented on the manuscript.

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
