[Supplementary Material]

# Supplementary

**Reconstruction results of the complete test cohort**

Figure 1: **Collage of reconstructions for the entire 'fMRI on ImageNet' test data (50 images).** Each pair of images shows the reconstructed image (right) side-by-side with its ground truth image (left).

Figure 2: **Collage of reconstructions for the entire 'vim-1' test data (120 images).** Each pair of images shows the reconstructed image (right) side-by-side with its ground truth image (left).

**Voxel selection**

To account for the differences in voxel reliability, we considered two measures. The first measure was voxel signal to noise ratio (SNR). By 'signal' we refer to the response variance when different images were presented, while 'noise' is defined as the variance across trials where the same image was presented. The second measure was reproducibility across repeated test trials (cf. intra-subject reproducibility in Wen et al.**?**). This refers to correlation of voxel response in a specific trial with the average response across trials; each series consists of 50 responses representing the number of distinct test images. We found full agreement between these two measures.

We accounted for voxel reliability distribution in our Decoder model by encouraging strong sparsity on the weights which are associated with low-SNR voxels. We used the inverse-SNR of each voxel to weigh the lasso regularization on the weights which are associated with it (weights from the fully-connected layer).

Figure 3: **Earlier visual areas have higher encoding prediction accuracy and higher SNR.**

**Impact of different repeat-counts of the test-fMRI**

A major factor underlying the performance leap when training on unlabeled test-fMRI (incorporating $\mathcal{L}^{DE}$) was the repeat-count of test-fMRI recordings. Averaging over multiple repeats results in fMRI samples with higher SNR, compared to those in the training data which only have a single repeat. Our ablation studies show that reconstruction improves as the number of averaged test-fMRI repeats increases (see Fig. 4 below). We considered the reconstruction results when applying our full method to subsampled-dataset versions with repeat count 1, 5, 10, 20, or 35.

This shows that our D-E architecture exploits the better SNR of the test-fMRI and adapts $D$ to the statistics of the test-fMRI, which deviates from the statistics (SNR) of the training data.

Figure 4: **Reconstruction quality gradually improves as the number of the averaged test-fMRI repeats per stimulus increases.** Randomly selected 1, 5, 10, 20, or 35 test-fMRI repeats were averaged and used for both unsupervised training and inference. Red frame indicates the configuration used in the paper.

**Spatial Grouping**

Our encoding and decoding models contain fully connected layers that constitute the transition between vector form voxel representation and spatial feature maps tensor. Each voxel had $\{w_{ij}^c\}$ weights connecting it to feature map $c$ at location $i, j$. Using this notation we define the spatial grouping loss per voxel,

$$\mathcal{L}_{SG} = \sum_{ij} \sqrt{\sum_c \kappa_{ij}^c} \tag{1}$$

$$\kappa_{ij}^c = (1-\alpha)\left(w_{ij}^c\right)^2 + \frac{\alpha}{4} \sum_{\Delta_i,\Delta_j \in \{-1,1\}} \left(w_{i+\Delta_i,j+\Delta_j}^c\right)^2, \tag{2}$$

where we used $\alpha = .5$ to force the grouping. Equation 1 expresses $L1$ (sparsity) loss in the spatial dimensions and an $L2$ loss in the channel dimension. This encourages an all-or-none inclusion of feature maps across spatial locations depending on their selection. The grouping force is due to the second term in Equation 2. Intuitively, this term accepts the weights of the neighboring locations at a reduced cost.

Note that the above regularization is a particular case of group lasso solution as defined in **?**.

where $\beta_j$ is a vector of the weights associated with a specific spatial location, denoted by $j$: these are the weights of that location and its four perpendicularly neighboring locations; $K_j$ is a diagonal positive definite matrix that we parameterize with $\alpha$.

**Loss components balancing**

The training loss of our Decoder is expressed by Eq. 2 in the paper:

$$\mathcal{L}^D + \mathcal{L}^{ED} + \mathcal{L}^{DE}.$$

The 3 components of the loss ($\mathcal{L}^D$, $\mathcal{L}^{ED}$, $\mathcal{L}^{DE}$) are normalized to have the same order of magnitude (all in the range $[0, 1]$), and same weights. This guarantees that no loss is dominated by the other two.

We applied two approaches to analyze the impact of the three components of the loss: (i) change the relative weights of the image and voxel reconstruction losses, or (ii) alter the ratio between the unlabeled and labeled examples in each batch. Fig. 5 below shows the effect of both of these – altering the ratio of unlabeled/labeled samples per batch (8, 16, or 32 unlabeled with 64 labeled samples), and assigning different weights $\times 2$ to the image/voxel losses. This shows that our reconstruction results are relatively insensitive to the exact balancing between the three components.

Figure 5: **Varying loss-weights & labeled/unlabeled ratio.** Red frame indicates the configuration used in the paper.