[Reviews · NeurIPS 2019]

Reviewer 1



The paper's writing and figures are of very high clarity and quality. The method is novel and the basic innovation is in the new objective function, which has encoder-decoder dynamics that are intriguing. The area of research is tackling the difficult problem of trying to reconstruct images from human brain activity with recent machine learning and neural network techniques, which is a strong fit for the NeurIPS conference. The results in Figure 4e) are impressive and look like a convincing improvement over Shen et al. 2019 as they do not need a generative model prior at all, but train an end-to-end architecture. The only ImageNet statistics in their network are pretrained low-level AlexNet features (thus also further lowering the potential influence of category set statistics). The fact that I am singling out the results in Fig. 4 e) as being impressive however brings me to the major shortcoming of this work: Target test data was used for training the decoder. I'm not convinced that the fact that there was no clear association between target fMRI and image resolves this problem. The reasons for this are that: * The test sets of the two used data sets consist of 50 and 120 images respectively. The L^DE loss being computed on test could essentially lead to learning to reconstruct the tiny set of test images from fMRI and fMRI from the reconstructed test images. It is unclear what is really being learned here. * Strong evidence for the truth of this statement is that 4e) has visibly diminished quality from the reconstructions where target test was included. It is likely that the model simply has learned by heart to encode(decode(testBOLD)). * The absolute magnitude of the two image losses will be very different from the L^DE loss. I wouldn't be surprised if this loss is dominating the image losses, further increasing the influence of learning to reconstruct the target test data. Also, the underlying motivation for tackling the problem of different train and test statistics in reconstruction data sets is difficult to grasp. Most of the data sets available in the field use test sets that are averaged over multiple repetitions to increase SNR, which indeed leaves the test set with different statistics, which indeed can have a diminishing influence when the aim is reconstruction. This has to be done for different reasons: Partly because the problem is too difficult with currently available brain imaging methodology without artificially reducing noise; partly because the aim of studies like Nishimoto 2011 was testing a certain hypothesis for brain representations (with the reconstruction's aim showing that the model is indeed powerful), needing high-SNR data to lead to correct results. (By the way, to readers unfamiliar with the field the paper should optimally explain why the field is using higher-SNR test data at all.) A general-purpose visual system decoder, which may be the ultimate aim of this research line should adapt to or not expect different statistics in training data and the data that should be reconstructed. So it is unclear whether adapting to different test set statistics is the most important problem to solve.

Reviewer 2



For decoding brain activity observed with fMRI, most methods require lots of stimulus-fMRI pairs. However, the experiments often yield insufficient data for end-to-end learning of the mapping from stimulus to fMRI or vice versa. This paper reports a clever idea, illustrated in Fig. 1, to train an encoder-decoder model with paired fMRI-stimulus data as well as unpaired or unlabeled fMRI or images. - The use of encoder-decoder for fMRI decoding is not new. For example, Han et al., (2019) uses a variational encoder to encode/decode natural images, while linearly mapping the latent variables to/from fMRI patterns. The encoder/decoder is also trained with unlabeled images. The authors' model is perhaps similar, but assumes the latent representation to be identical to the fMRI patterns. - What is novel here is that the fMRI data (without being paired to images) can be used to train the encoder/decoder as well, but reversing the direction of the computational graph. This is clever and inspiring. However, the results do not show any compelling advantage. It is sometimes hard to tell which methods are better based on the reconstructed images, which are nearly equally poor. - Although the method is interesting and has novelty, the neuroscientific insight is limited.

Reviewer 3



There are many choices of hyper parameters, such as alpha, or the choice of the architecture. How to safegard against overfit of the test metric? Given that the test sets are of limited size, it risks happening fast. With regards to the encoding architecture, starting from the images, I would be curious to see how unsupervised training compares to classic pretrained architectures. In equation 3, it is not obvious to me how to balance the three terms of the loss: how it is done in practice, and what are the guiding principles.

Reviewer 4



The authors propose a pipeline for decoding natural images from fMRI recordings. Their specific contributions is clever way to overcome the scarcity of training data in this setting, e.g., of pairs {images,fMRI recordings}. Specifically, they show how to adapt their model in an unsupervised fashion on natural images only and on (test) fMRI data only. They carefully evaluate the performance of their model and compare with the state-of-the-art. This is one of the rare manuscripts where I really have nothing to criticize. The paper is very well written, the proposed model is novel and clever, and the obtained results are (slightly above the) state-of-the-art and critically examined. In my opinion, this is as good as fMRI decoding currently gets, and a clear accept.

[Author Response · NeurIPS 2019]

# From voxels to pixels and back: Self-supervision in natural-image reconstruction from fMRI

We thank the reviewers for their comments and endorsements. Below are our answers to the main questions/concerns.

**R1: Training on test-fMRI samples – not convinced the approach is valid.** We understand the reviewer's concern. Note however that our "training on test data" refers only to training on ***unlabeled samples from the Decoder's input space** (test-fMRI)*, whereas the *test-images* (the "labels") are never used at any stage of the training. Thus, such training is valid. We will better clarify the distinction between training on the "test-fMRI" (which is the input to the network, hence totally valid to train on), vs. training on the the "test-images" (which is the desired output of the network, hence illegal/invalid to train on, and indeed we do not). We realize that this distinction is confusing, and will clarify it.

**R1: A general-purpose Decoder is the ultimate aim.** We respectfully beg to differ. We believe that a 'universal' fixed Decoder for arbitrary input fMRI samples is inconceivable given the limited training data, and the large variability in fMRI acquisition parameters and subjects. Instead, we propose an adaptive decoder that learns to adapt to new fMRI samples (of never-before-seen images, and with new statistics), as they come along. This adaptation underlies the high performance of our Decoder on new fMRI data. Since this holds for any set of test-fMRI (which can be incorporated in training) our self-supervised Decoder has significant capability to generalize well to new/held-out data.

**R1: Different statistics of fMRI in the Train & Test datasets.** Indeed, the train/test fMRI SNR discrepancy results from averaging a different number of repeated recordings per image (typical of many fMRI datasets). This statistical discrepancy introduces an additional challenge of 'domain transfer/adaptation', affecting the performance of current decoding methods. Our $\mathcal{L}^{DE}$ objective directly addresses this issue: It enables to learn the *different* statistics of the test-fMRI, and specifically of the target-fMRI. Fig. A shows the impact of a different number of repeat-counts of the test-fMRI (averaging 1, 5, 10, 20, or 35 randomly selected repeats). Reconstruction improves as the number of test repeats (SNR) increases *in the Test-set only* (Train-set remains the same). It shows that our D-E architecture exploits the better SNR of the test-fMRI and adapts $D$ to the statistics of the test-data. We will add this explanation/figure to the paper/supp-material. As suggested by R1, we will add more background regarding SNR/repeat-count.

**R1+R4: Loss magnitudes & Loss ablation experiments.** The 3 components of the loss in Eq.2 ($\mathcal{L}^D$, $\mathcal{L}^{ED}$, $\mathcal{L}^{DE}$) are *normalized* to have the same order of magnitude (all in the range $[0, 1]$). This guarantees that no loss is dominated by the other two. We therefore assigned equal weights to all losses. We will add these clarifications. There are two simple ways to examine the relative effect of the different loss terms: (i) change the relative weights of the image and voxel reconstruction losses, or (ii) alter the ratio between the unlabeled and labeled examples in each batch. Fig. B shows the effect of both of these – altering the number of unlabeled examples per batch (8, 16, or 32 samples out of 64), and assigning different weights $\times 2$ to the image/voxel losses. We will add this explanation+figure to paper/Supp-Material.

**R4: Choice of hyperparameters; potential risk of overfitting the test dataset – Maybe try another dataset?** This is an important point which we failed to stress: Our method was applied on two separate (and very different) datasets using *the exact same* hyperparameters. We will add this important clarification to the paper. This is strong evidence to the general applicability of our method to very different datasets with very different statistics. We are not aware of any other publicly available fMRI dataset of *natural images* for which there are reported reconstructions. We found our reconstructions to be relatively robust to variations in the hyperparameters (as can also be seen in Fig. B). We further performed a cross-validation procedure for selecting the 'optimal' set of hyperparameters based on the train-set alone. This yields very similar hyperparameters to those we used, and similar reconstructions.

**R4: Neuroscientific insights.** While we believe that image-reconstruction can eventually become a strong neuroscientific tool, this is not the focus of the current paper. This paper highlights a new learning method/approach, which is exemplified on a difficult neuroscientific problem, but is not limited to it. We believe this approach may apply to other domains and problems, which are characterized by scarce labeled data.

Figure A: Effect of averaging repeated fMRI recordings. Figure B: Varying loss-weights & labelled/unlabeled ratio.

[Meta-Review · NeurIPS 2019]

Dear authors, congrats on the acceptance-- this paper was discussed extensively, the the reviewers provided multiple comments and feedback-- please do take the feedback and requests of all the reviewers into account when preparing your final manuscript. In particular, it would be important to clearly describe in what settings (i.e. what labelled and unlabeled data needs to be available at test time) the proposed approach can yield benefits over baselines.